# Reverse Water-Gas Shift Iron Catalyst Derived from Magnetite

Chen-Yu Chou, Jason A. Loiland and Raul F. Lobo *

Center for Catalytic Science and Technology, Department of Chemical and Biomolecular Engineering, University of Delaware, Newark, DE 19716, USA; cychou@udel.edu (C.-Y.C.); jasonloiland@gmail.com (J.A.L.)
* Correspondence: lobo@udel.edu; Tel.: +1-302-831-1261

**Abstract:** The catalytic properties of unsupported iron oxides, specifically magnetite ($Fe_3O_4$), were investigated for the reverse water-gas shift (RWGS) reaction at temperatures between 723 K and 773 K and atmospheric pressure. This catalyst exhibited a fast catalytic CO formation rate (35.1 mmol $h^{-1}$ $g_{cat.}^{-1}$), high turnover frequency (0.180 $s^{-1}$), high CO selectivity (>99%), and high stability (753 K, 45000 $cm^3h^{-1}g_{cat.}^{-1}$) under a 1:1 $H_2$ to $CO_2$ ratio. Reaction rates over the $Fe_3O_4$ catalyst displayed a strong dependence on $H_2$ partial pressure (reaction order of ~0.8) and a weaker dependence on $CO_2$ partial pressure (reaction order of 0.33) under an equimolar flow of both reactants. X-ray powder diffraction patterns and XPS spectra reveal that the bulk composition and structure of the post-reaction catalyst was formed mostly of metallic Fe and $Fe_3C$, while the surface contained $Fe^{2+}$, $Fe^{3+}$, metallic Fe and $Fe_3C$. Catalyst tests on pure $Fe_3C$ (iron carbide) suggest that $Fe_3C$ is not an effective catalyst for this reaction at the conditions investigated. Gas-switching experiments ($CO_2$ or $H_2$) indicated that a redox mechanism is the predominant reaction pathway.

**Keywords:** RWGS; iron oxides; $CO_2$ conversion; gas-switching

## 1. Introduction

Today's anthropogenic emissions of carbon dioxide to the atmosphere amount to about 35,000 Tg per year, and the greenhouse effect of these accumulated emissions has been recognized as an alarming hazard to the well-being of modern societies. Although multiple approaches have been considered to mitigate these emissions, recently more emphasis has been placed on the potential synergy of carbon capture and the utilization of these large outflows of carbon dioxide. Among several possibilities, a recent National Academies of Science and Engineering report [1] on $CO_2$ waste gas utilization highlights the conversion of $CO_2$ by hydrogenation into CO and water—the reverse water-gas shift (RWGS) reaction, Equation (1)— as critical, and points to the need for improved catalysts with high stability and durability.

The RWGS reaction can be part of a two-step hydrogenation process for the conversion of $CO_2$ to valuable products. First, $CO_2$ is reduced to CO via the RWGS reaction, and second, CO can be converted to either hydrocarbons via the Fischer-Tropsch (FT) process or methanol via CAMERE (CArbon dioxide hydrogenation to form MEthanol via a Reverse WGS reaction) process [2]. The RWGS reaction is an endothermic reaction ($\Delta H°_{298 K}$ = 41.2 kJ $mol^{-1}$), and thus is thermodynamically favorable at higher temperatures.

$$CO_2 + H_2 \leftrightarrow CO + H_2O \tag{1}$$

Noble metals such as Pt [3–5] and Pd [6,7], and other various metals such as Cu [8–11], Ni [12,13] and Fe [14] supported on oxides were reported to be active for the production of CO. Among them, Cu-based materials have been widely studied, and thus have also been investigated in many instances

for the RWGS reaction. For example, Cu-Ni/Al$_2$O$_3$ [15], Cu/ZnO [16], Cu-Zn/Al$_2$O$_3$ [16] and Cu/SiO$_2$ promoted with potassium [17] have all shown good RWGS activity. However, Cu materials tend to deactivate by sintering (frittage) at high temperatures (T > 773 K), which are required for high RWGS activity. For high temperature applications, iron can be added as a thermal stabilizer: Chen et al. [10] showed that adding small amounts of iron to 10% Cu/SiO$_2$ resulted in stable RWGS activity for 120 h at 873 K and atmospheric pressure, while non-promoted 10% Cu/SiO$_2$ deactivated rapidly.

Iron oxides (Fe$_x$O$_y$) are often used industrially for FT synthesis (473 K–623 K, 1 MPa) [18,19] and the high-temperature (623 K–723 K) WGS reaction [20–22]. In FT synthesis, alkalized, iron-based catalysts are combined with Cu for reduction promotion. Schulz et al. [18,23] showed that the working FT catalysts contain several iron carbide phases and elemental carbon formed after the hydrogen reduction period. Iron oxides and the metallic iron are still present in the catalyst composition, but iron carbide phases are identified as active sites [23]. The RWGS and WGS reactions are often carried out in conjunction with FT synthesis at a higher temperature regime on iron catalysts, and iron oxide or oxidic amorphous iron phases are known as the active phases for WGS and RWGS [24–26].

Extensive research on iron-based catalysts has been reported mainly on the WGS reaction over decades [20]. Chromium is a structural promoter that helps prevent the iron from sintering at high temperature. A more recent survey on Cr-free, Fe-based WGS catalysts shows the current strong interest in this topic [27]. However, the studies on RWGS reactions over iron-based catalysts are much less frequent. Fishman et al. [28] synthesized hematite nano-sheets to obtain a 28% CO$_2$ conversion at 783 K, and hematite nanowires to obtain a 50% CO$_2$ conversion at a very high temperature of 1023 K. Hematite was reduced to magnetite during the reaction. The catalytic behavior over time on stream and the stability of the CO production were, however, not investigated. Fe nanoparticles have also shown good stability and activity (35% CO$_2$ conversion, >85% CO selectivity) in RWGS by Kim et al. [29], yet no kinetic parameters were determined, and the mechanism was not discussed.

Two principal mechanisms of the WGS (or RWGS) reaction have been investigated extensively: The "redox mechanism" and the "associative" mechanism [30,31]. Different catalysts may lead to a different reaction pathway. The redox mechanism was suggested to be active for the WGS reaction over iron catalysts promoted with chromium [32]. A distinguishing feature of the redox mechanism is that products can be generated in the absence of both reactants. The catalyst is first reduced by the adsorbed H$_2$, and is subsequently oxidized by CO$_2$ (in RWGS) or H$_2$O (in WGS). The associative mechanism was proposed to be dominant in the WGS reaction over iron oxide catalysts [33]. In this mechanism, both reactants are adsorbed on the catalyst surface at the same time to create products. Several carbon-containing intermediates, including formate, carbonate, carbonyl and carboxyl species, have been proposed. In a previous report in alumina-supported iron catalysts [14], we showed that the redox mechanism is the only pathway for RWGS over Fe/γ-Al$_2$O$_3$, and the predominant pathway over Fe-K/γ-Al$_2$O$_3$. The addition of the potassium promoter activates a secondary pathway for CO formation, which is probably the associative pathway.

In the present report, unsupported Fe$_3$O$_4$-derived catalyst is identified as a highly active, selective and stable catalyst for the reverse water-gas shift reaction at temperatures between 723 K and 753 K. The characterization of surface composition, bulk properties, and the evaluation of the CO production specific rate showed that the working catalyst is constructed during the H$_2$-activation and the period of reaction conditions. Quantitative gas-switching experiments in combination with isotopic switching experiments allowed the redox and associative reaction pathway to be differentiated. The catalysts appear to be highly stable under the reaction conditions investigated.

## 2. Results and Discussion

The catalytic CO formation rates on the Fe$_3$O$_4$ catalyst with various H$_2$ to CO$_2$ ratios (Figure 1) show that after an induction period of ~120 min, the catalyst produced CO at a steady rate of 35.1 mmol h$^{-1}$ g$_{cat.}$$^{-1}$ at 753 K with 12.5% CO$_2$ conversion. The selectivity to CO was greater than 99% under equimolar CO$_2$ and H$_2$. After 950 min, the partial pressure of CO$_2$ was raised to 60 kPa,

while the partial pressure of $H_2$ was kept constant. The rate increased to 54.6 mmol $h^{-1}$ $g_{cat.}^{-1}$ with 4.4% $CO_2$ conversion. Deactivation also occurred during this period: Starting at a deactivation rate of 3.71 mmol $h^{-1}$ $g_{cat.}^{-1}$ per h, this rate gradually decreased to 0.23 mmol $h^{-1}$ $g_{cat.}^{-1}$ per h. When the partial pressure of $H_2$ was 60 kPa, and the $CO_2$ was switched back to 15 kPa, the CO formation rate increased first to 91.3 mmol $h^{-1}$ $g_{cat.}^{-1}$ and gradually stabilized to a value of 95.3 mmol $h^{-1}$ $g_{cat.}^{-1}$ with 33.7% $CO_2$ conversion. The final rate of the catalyst reactivation was about 0.50 mmol $h^{-1}$ $g_{cat.}^{-1}$ per h. It should be noted that the differential condition (see Equation (8)) was used to determine reaction rates, and preferred for the investigation of the kinetic properties of our materials. Under higher concentration of $H_2$ (15 kPa $CO_2$ + 60 kPa $H_2$), high $CO_2$ conversion (>12%) could lead to small errors in the estimation of the reaction rate and the reaction order. The trend observed in Figure 1, however, should not be affected by this approximation.

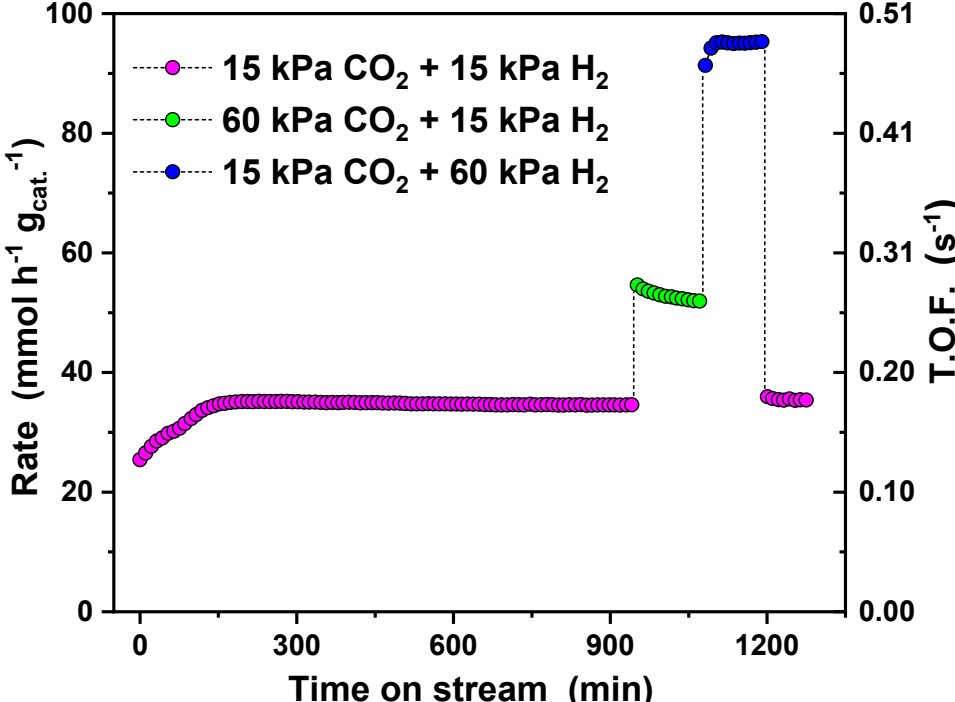

**Figure 1.** CO formation rates and their turnover frequencies (T.O.F.) on $Fe_3O_4$ at partial pressures of $CO_2$ and $H_2$ indicated in the legend. Other reaction conditions: $P_{tot}$ = 1 bar, T = 753 K, $F_{tot.}$ = 75 sccm, GHSV = $4.5 \times 10^4$ $cm^3$ $h^{-1}$ $g_{cat.}^{-1}$.

With 60 kPa of $H_2$ and 15 kPa of $CO_2$ in the feed, a small amount of $CH_4$—the only side-product—was produced at the rate of 1.35 mmol $h^{-1}$ $g_{cat.}^{-1}$, reducing the CO selectivity from near 100% to 98.6%. Methane production implies that C–H bond formation is facilitated at higher partial pressure of $H_2$. There was no further C–C chain growth under this reaction condition, indicating that the FT synthesis was not active over the working catalyst. The effect of $H_2$ partial pressure on the CO formation rate was much higher than the effect of $CO_2$, implying a higher reaction order on $H_2$ than $CO_2$. The catalyst showed overall high stability in 1300 min, and the final reactivation rate in excess $H_2$ was higher than the deactivation rate under the excess $CO_2$ condition. This indicates that this $Fe_3O_4$-derived catalyst is easy to regenerate in a very short period (<100 min), making the catalyst attractive in industrial use for long-term application.

The activation of the catalyst was carried out by a pretreatment under reducing conditions ($H_2$ gas). The bulk structure of the catalyst after the pretreatment can be identified. Based on the body-centered cubic (BCC) structure of α-Fe (JCPDS PDF 00-006-0696) after the pretreatment (Figure S1), the surface density of Fe atoms on the Fe(110) surface can be calculated as $1.297 \times 10^{19}$ Fe atoms $m^{-2}$. Assuming

all Fe atoms on Fe(110) were active sites, the observed CO formation rates can be converted to turnover frequency (TOF). This is reported in Figure 1 based on the measured CO formation rates, atomic surface density, and BET surface area (2.52 $m^2/g$ for $Fe_3O_4$) of the pristine catalyst. The turnover frequency of this catalyst under the equimolar condition was as high as 0.18 $s^{-1}$ ($P_{tot}$ = 1 bar, T = 753 K, $F_{tot.}$ = 75 sccm, GHSV = $4.5 \times 10^4$ $cm^3$ $h^{-1}$ $g_{cat.}^{-1}$).

The stable reaction rates observed after the initial break-in period at 753 K allow for the determination of kinetic parameters without having to model deactivation profiles. The kinetic parameters, including reaction orders with respect to $CO_2$ and $H_2$, and measured activation energies ($E_{meas}$) over $Fe_3O_4$ under near equimolar $CO_2$ and $H_2$ (~1:1), and in $H_2$ excess (2:1, 4:1, and 9:1)—see Table 1— indicate that CO formation rates have a higher dependence on $H_2$ partial pressure (order of ~0.8) than $CO_2$ partial pressure (order of ~0.33) under equimolar composition. In general, rate orders depend on reaction conditions: Increasing the $H_2$ partial pressure increases the order on $CO_2$ to 0.39 and decreases the rate order on $H_2$ to 0.72. At a ratio of $H_2$:$CO_2$ near 4:1, the reaction orders still show the same trend: An increasing dependence on $CO_2$ (order of 0.43) and decreasing dependence on $H_2$ (order of 0.31). In a high excess of $H_2$ ($H_2$:$CO_2$ = ~9:1), the reaction rate over $Fe_3O_4$ was of the order 1.30 with respect to $CO_2$, and was independent of $H_2$ pressure. The activation energies ($E_{meas}$) also depend on the $H_2$:$CO_2$ partial pressure ratios; that is, different reaction pathways may occur under these conditions. This behavior is not unique to iron catalysts. Similar reaction orders were also observed by Ginés et al. [34] in the same regime of $P_{H2}/P_{CO2} < 3$ ($CO_2$ order ≈ 0.3, $H_2$ order ≈ 0.8) on the CuO/ZnO/$Al_2O_3$ catalyst, and by Kim et al. [3] on Pt/$Al_2O_3$ catalysts ($CO_2$ order = 0.32, $H_2$ order = 0.70). It was also suggested by Ginés et al. [34] that different reaction pathways should be existed for $P_{H2}/P_{CO2} < 3$ and $P_{H2}/P_{CO2} > 3$ regions.

**Table 1.** Measured reaction orders with respect to $CO_2$ and $H_2$, and measured activation energies ($E_{meas}$) over $Fe_3O_4$. Reaction conditions: 100 mg $Fe_3O_4$, $F_{tot}$ = 75 sccm, T = 723 K.

| $P_{H2}$ (kPa) | $P_{CO2}$ (kPa) | $P_{H2}$: $P_{CO2}$ | Reaction Order in $CO_2$ | Reaction Order in $H_2$ | $E_{meas}$ (kJ/mol) |
|---|---|---|---|---|---|
| 15 | 10–20 | ~1:1 | 0.33 | - | 28.9 ± 0.9 |
| 10–20 | 15 | | - | 0.79 | |
| 30 | 10–20 | ~2:1 | 0.39 | - | 27.1 ± 0.5 |
| 25–35 | 15 | | - | 0.72 | |
| 40 | 5–12.5 | ~4:1 | 0.43 | - | 34.2 ± 1.9 |
| 35–45 | 10 | | - | 0.31 | |
| 85 | 5–12.5 | ~9:1 | 1.30 | - | 39.0 ± 3.4 |
| 70–90 | 10 | | - | 0 | |

Figure 2 presents the temperature-programmed reduction (TPR) profiles of (a) the fresh $Fe_3O_4$ sample and (b) the post-reaction $Fe_3O_4$ sample. In Figure 2a, the fresh $Fe_3O_4$ was heated to 753 K in a hydrogen atmosphere, kept for 2 h at these conditions, and then heated to 1,073 K at the rate of 5 K/min. The small peak observed in the TPR trace at about 563 K is assigned to an impurity of hematite present in the initial sample of magnetite ($Fe_3O_4$), but not detected in the XRD pattern, as shown in the report by Jozwiak et al. [35]. The following broader and asymmetric peak suggests a two-step reduction process that has been previously postulated in literature [36], as the following: (1) $Fe_3O_4 \overset{H_2}{\rightarrow} FeO$ and (2) $FeO \overset{H_2}{\rightarrow} Fe^0$. These two steps can be deconvoluted into two peaks located at ~ 688 K and 773 K in the TPR traces. After the 2 h reduction period at 773 K, there was no further $H_2$ consumption at higher temperatures. That is, the sample, after the reduction pretreatment used in our activation protocol, has been converted into metallic iron. This result is also consistent with the XRD pattern in Figure S1, which shows that $\alpha$-Fe was the crystal formed after the reduction pretreatment of the $Fe_3O_4$ sample in the microreactor.

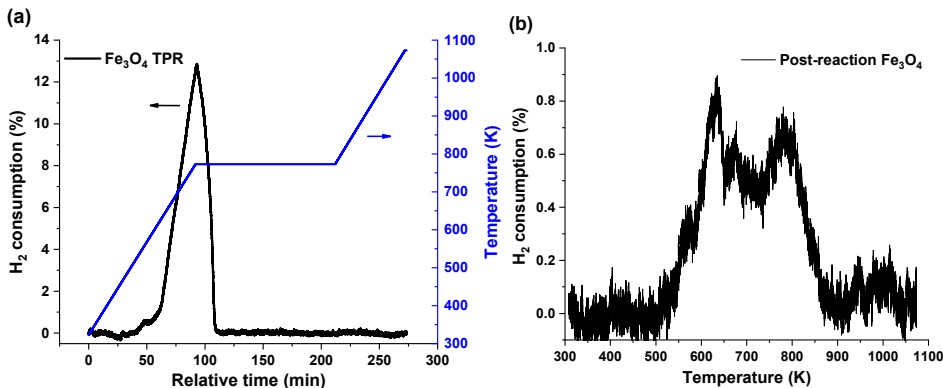

**Figure 2.** (**a**) $H_2$ consumption (%) of the reduction period during the pretreatment following the temperature-programmed reduction (TPR) of 100 mg $Fe_3O_4$. (**b**) TPR curve of the post-reaction $Fe_3O_4$ sample.

As soon as $CO_2$ was fed into the reactor, the surface of the catalyst was partially oxidized. This is known from the results of Figure 2b that illustrates the presence of two significant peaks in the TPR profile for the post-reaction catalysts (at 630 K and 780 K, respectively). The location of these two predominant peaks shows good agreement with the results of Figure 2a and results reported elsewhere [35,37]. There is at least a two-step reduction at ~630 K and 780 K, implying the coexistence of different oxidation states of iron on the post-reaction sample due to the partial oxidation from $CO_2$. For this post-reaction sample, the $H_2$ concentration in the effluent stream decreased by only a small amount (less than 1%), suggesting that the consumption of the $H_2$ feed would not affect the reduction rate during the TPR reaction.

Figure 3 displays the diffraction pattern of fresh $Fe_3O_4$ and the catalyst after the RWGS reaction (post-reaction $Fe_3O_4$). The 2θ degree peak positions in fresh $Fe_3O_4$ were 30.15°, 35.45°, 37.15°,43.15°, 53.50°, 56.95° and 62.55°, which are all consistent with magnetite (JCPDS PDF 01-071-6336). The post-reaction $Fe_3O_4$ shows a very different XRD pattern: This pattern was composed of metallic iron (α-Fe, 44.67°, shown in the inset of Figure 3), iron carbide ($Fe_3C$), and a small peak of $FeO_X$ (35.47°).

Iron oxides can be converted directly into carbides in a reducing and carburizing atmosphere [38], and the carbon source of the $Fe_3C$ production can be either from impurities in the fresh $Fe_3O_4$ sample or due to reaction with the product CO, after the RWGS reaction as indicated by Equation (2) [39]:

$$3Fe + 2CO \leftrightarrow Fe_3C + CO_2 \qquad (2)$$

The bulk composition of the catalyst after the reaction is also consistent with the TPR results in Figure 2a,b. During TPR, the amount of $H_2$ consumption of magnetite relative to the amount of $H_2$ consumption after the reaction was 12:1, therefore the iron species in the catalyst has been changed into a more reduced chemical state after the pretreatment and RWGS reaction. The reduction was mainly caused by the pretreatment, while the following $CO_2/H_2$ reaction shifted the metallic iron back to a slightly more oxidized state; a combination of iron carbide, metallic iron and some iron oxides.

Besides bulk property information obtained from XRD, XPS analyses were conducted to characterize the surface composition of the initial fresh $Fe_3O_4$ and the change of the catalyst after the RWGS reaction. Figure 4 shows the XPS spectra, peak deconvolutions and the fitting envelopes for the Fe $2p_{3/2}$ spectra of $Fe_3O_4$ and post-reaction $Fe_3O_4$.

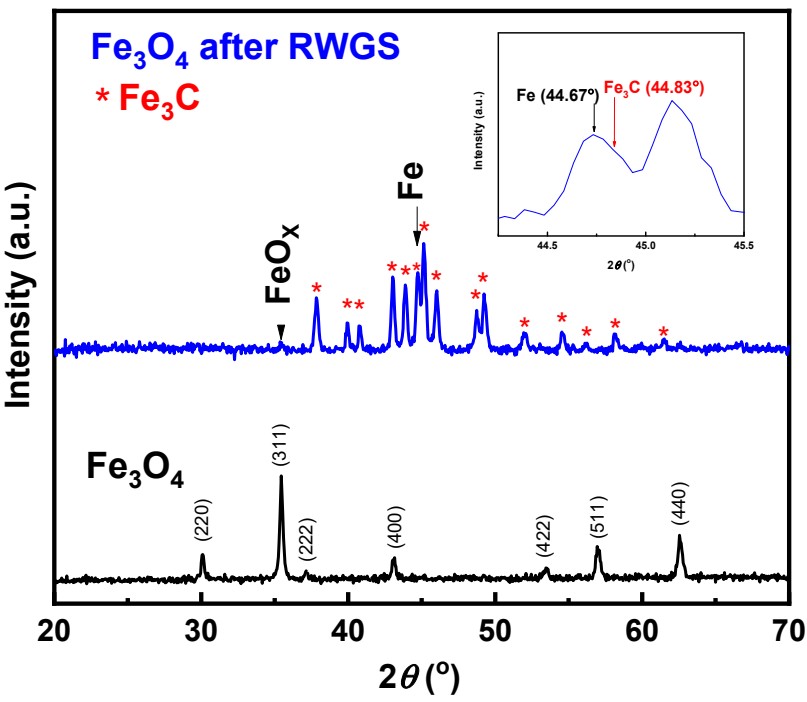

**Figure 3.** XRD patterns of fresh $Fe_3O_4$ (down) and post-reaction $Fe_3O_4$ (up); inset is the magnification of post-reaction $Fe_3O_4$ from 44° to 45.5°.

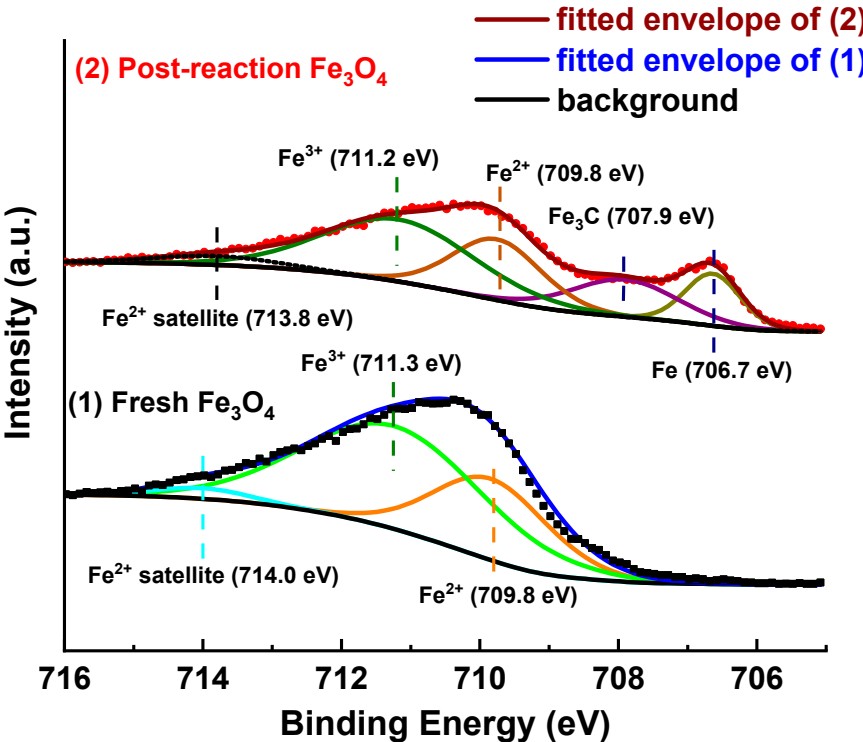

**Figure 4.** XPS Fe $2p_{3/2}$ spectra of $Fe_3O_4$ and post-reaction $Fe_3O_4$. The curves under the fitted envelope and above the background are contributions of estimated components from peak fitting.

Atomic percent contributions are calculated from the fitted peaks of Fe $2p_{3/2}$ due to its larger intensity (Area of Fe $2p_{3/2}$:Fe $2p_{1/2}$ = 2:1). The Fe $2p_{3/2}$ spectra were fitted over the range of 705–722 eV. The spectra between 716–722 eV were not shown in the figure for clarity. In this range, there were only

small $Fe^{3+}$ satellite peaks located at 719.2 eV and 719.4 eV for both the fresh and post-reaction $Fe_3O_4$ samples, respectively, although the area of the satellite peaks was still included in the corresponding components when calculating the relative atomic percentage. The fits, including the binding energy, full width at half maximum (FWHM), and the relative iron composition, are summarized in Table 2. The fitted XPS spectrum of fresh $Fe_3O_4$ was composed of doublets for $Fe^{2+}$ at 709.8 eV and $Fe^{3+}$ at 711.3 eV. $Fe_3O_4$ has an inverse spinel structure which can be written as $Fe^{3+}_{TET}[Fe^{2+}Fe^{3+}]_{OCT}O_4$, with one $Fe^{3+}$ on a tetrahedral site, and $Fe^{2+}$ and the other $Fe^{3+}$ distributed on octahedral sites. Therefore, the theoretical relative composition of $Fe^{2+}/Fe^{3+}$ is 0.5, which is close to the area fitted and the calculated relative composition in our fresh $Fe_3O_4$ ($Fe^{2+}:Fe^{3+}$ = 34.8/65.2 = 0.53). The $Fe^{3+}$ peak has a larger FWHM than $Fe^{2+}$. This is as expected because the electronic configuration of $Fe^{2+}$ is $3d^6$, while that of $Fe^{3+}$ is $3d^5$, that is, $Fe^{2+}$ will have a longer life time compared to $Fe^{3+}$; and therefore the FWHM of the $Fe^{2+}$ peak should be smaller than the $Fe^{3+}$ peak [40]. Additionally, the $Fe^{3+}$ peaks can be attributed to two different structures, octahedral $Fe^{3+}$ and tetrahedral $Fe^{3+}$, a factor that will also lead to broader $Fe^{3+}$ peaks.

**Table 2.** XPS peak fittings for Fe $2p_{3/2}$ spectra for $Fe_3O_4$ and post-reaction $Fe_3O_4$.

| Sample | Peak Deconvolution | Binding Energy (eV) | FWHM (eV) | Atomic % |
|---|---|---|---|---|
| $Fe_3O_4$ | $Fe^{3+}$ | 711.3 | 3.08 | 65.2 |
|  | $Fe^{2+}$ | 709.8 | 1.95 | 34.8 |
| post-reaction $Fe_3O_4$ | $Fe^{3+}$ | 711.2 | 2.58 | 43.9 |
|  | $Fe^{2+}$ | 709.8 | 1.51 | 27.5 |
|  | $Fe_3C$ | 707.9 | 1.78 | 16.4 |
|  | Metallic Fe | 706.7 | 0.94 | 12.2 |

After the RWGS reaction, the spectrum was fitted using four different components corresponding to metallic Fe (706.7 eV), $Fe_3C$ (707.9 eV), $Fe^{2+}$ (709.8 eV) and $Fe^{3+}$ (711.2 eV). The peak locations of $Fe^{2+}$ and $Fe^{3+}$ were the same or very close to the fresh sample, indicating that there was only a small surface charging effect with the flood gun on. The binding energies of the components are in agreement with literature results [38,40–42]. In terms of the atomic percentages, the overall peak area was re-allocated to a more reduced regime after the RWGS reaction. The $Fe^{2+}$ decreased from 34.8% to 27.5%, the $Fe^{3+}$ decreased from 65.2% to 43.9%, while there were two components formed: Fe (12.2%) and $Fe_3C$ (16.4%). The shift of the spectra was due to the $H_2$ reduction pretreatment before operating the RWGS reaction, and the flow of $H_2/CO_2$ reactants through the system would balance each other to make the catalyst partially oxidized or reduced. Though the sample surface could be oxidized by the air during the transportation from the reactor to the XPS analysis chamber, the result from XPS still can confirm the reduction of the surface during the reaction, since new crystal structures such as Fe and $Fe_3C$ are detected by XRD.

The XPS analyses indicate that the active catalyst consisted of a mixture of metallic Fe, $Fe_3C$, $Fe^{2+}$ and $Fe^{3+}$; however, it cannot establish the relative contributions of these components to the observed rate of the RWGS reaction. To determine if the iron carbide formed in our reaction can catalyze the RWGS reaction, reaction rates over pure $Fe_3C$ were measured (Figure 5). In Figure 5a, $Fe_3C$ showed an initial CO formation rate of 26.0 mmol $h^{-1}$ $g^{-1}$, but dropped 48% to near 13.5 mmol $h^{-1}$ $g^{-1}$ in 10 min, and then further down to 5.7 mmol $h^{-1}$ $g^{-1}$ in 160 min at 753 K. This is clearly different from the properties of the $Fe_3O_4$-derived catalyst in Figure 1, since the $Fe_3O_4$-derived catalyst displayed high stability for at least 1300 min. To evaluate the fast deactivation shown in Figure 5a and remove the initial reduction effect during the ramping by hydrogen, another measurement was carried out with respect to temperature, (see Figure 5b). This experiment was conducted flowing a $CO_2/H_2/He$ gas mixture in the same relative concentration used in the standard activity test during the ramping procedure from room temperature to 773 K. At 573 K, the material did not catalyze the formation

of CO; however, when the temperature was increased to 623 K, the catalyst immediately showed catalytic rates in the range of 2.60 mmol $h^{-1}$ $g^{-1}$ to 2.99 mmol $h^{-1}$ $g^{-1}$. After 1 h of the reaction, the rate remained at quasi-steady state at this low temperature (623 K).

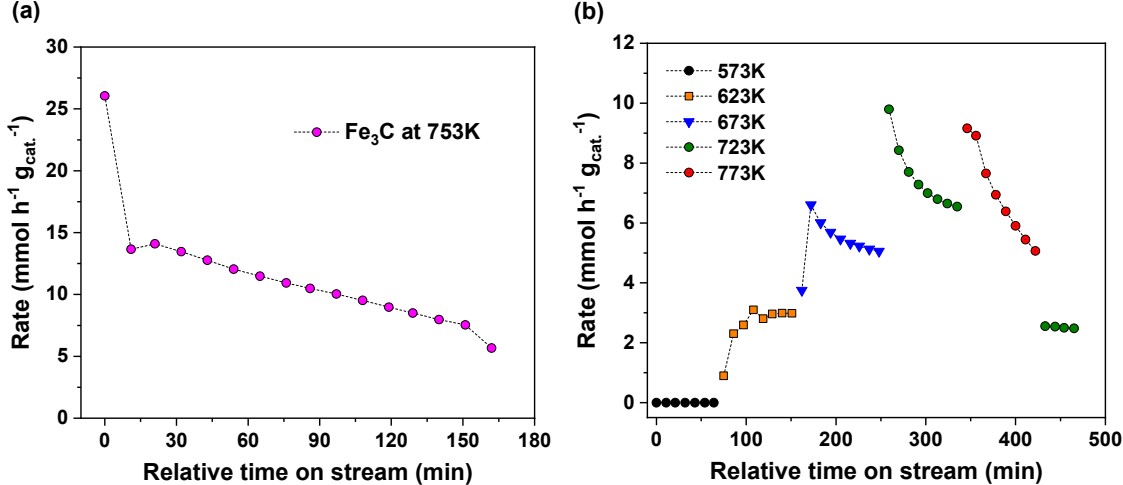

**Figure 5.** (**a**) CO formation rate on 100 mg $Fe_3C$. Reaction conditions: $F_{tot}$ = 75 sccm; the reactor was ramped to 753 K with $P_{H2}$ = 15 kPa and He as remainder. No further reduction was applied after the ramping. During the reaction, T = 753 K, $P_{tot}$ = 1 bar, $P_{H2}$ = $P_{CO2}$ = 15 kPa. (**b**) CO formation rate on 100 mg $Fe_3C$. The temperature was directly ramped from room temperature to 773 K as shown in the figure with $P_{tot}$ = 1 bar, $P_{H2}$ = $P_{CO2}$ = 15 kPa, and the remainder is He.

The CO formation rates were 6.60 mmol $h^{-1}$ $g^{-1}$, 9.80 mmol $h^{-1}$ $g^{-1}$, and 9.16 mmol $h^{-1}$ $g^{-1}$, as the temperature was increased to 673 K, 723 K and 773 K, respectively. The formation rate at 773 K was not higher than that at the lower temperature because of rapid deactivation at this temperature. Faster deactivation rates were observed at higher temperatures: The average deactivation rates were 1.36 mmol $h^{-1}$ $g^{-1}$ per h, 2.57 mmol $h^{-1}$ $g^{-1}$ per h and 3.50 mmol $h^{-1}$ $g^{-1}$ per h.

After reacting at 773 K, the temperature was reduced to 723 K to monitor the reaction rate and compare to the previous value. Much lower rates (2.55 mmol $h^{-1}$ $g^{-1}$) were observed than in the previous measurement at the same temperature (723 K), indicating that there was an irreversible change in the catalyst structure or composition. At higher temperatures, the reverse reaction of Equation (2), where $Fe_3C$ reacts with $CO_2$ and forms Fe and CO, is more favorable than the $Fe_3C$ formation [39]. Therefore, the initial CO formation rate was probably due to the formation of CO from decomposition, but quickly dropped since it is harder to convert the metallic iron back to $Fe_3C$ at higher temperatures. The iron carbide catalyst only showed steady CO production at 623 K. This explains why the deactivation at 723 K over $Fe_3C$ is very different from the steady-state magnetite catalyst reported in Figure 1, which showed a very stable CO formation rate at 753 K. This observation suggests that the operating temperature of RWGS in this study was not an environment conducive for a stable iron carbide for CO production. In addition, the iron carbide ($Fe_5C_2$ or $Fe_3C$) is normally considered to be the active phase of iron for hydrocarbon production [43,44], and iron oxide is the active phase for WGS and RWGS [25]. Several reports have suggested that the stability of the iron catalyst in either FT synthesis [45] or RWGS [29] can be related to an iron carbide layer. Davis [45] suggested that catalyst composition and reaction condition will define the existence of the pseudo-equilibrium layer of iron carbide to ensure a very slow deactivation condition. Kim et al. [29] concluded that the stability of the catalyst could have originated from migration of C and O into the catalyst bulk, forming iron oxide and iron carbide, which likely prevented the nanoparticles on the surface from agglomerating. Based on the XPS results and $Fe_3C$ catalytic tests, the iron carbide of the working catalyst is less likely to be the main active site for CO production, but is an important species to provide stability in the overall catalytic performance.

Gas-switching experiments, in which $H_2$ and $CO_2$ are flown on and off, were used to distinguish and quantify contributions from redox and associative reaction pathways [3,14,46]. In the simplest form of the redox mechanism, gas-phase $CO_2$ adsorbs on a reduced site to form CO and an oxidized site (Equation (3)), which can then be reduced by gas phase $H_2$ to reform the reduced site (Equation (4)). The simplest redox cycle can be described as follows:

$$CO_2(g) + s_{red.} \rightarrow CO(g) + O \cdot s \tag{3}$$

$$H_2 + O \cdot s \rightarrow H_2O(g) + s_{red.} \tag{4}$$

A simplified associative pathway can be described generally by Equation (5). $CO_2$ and $H_2$ adsorb on the catalyst surface to form a carbon-containing intermediate (i.e. formate, carbonate, or bicarbonate), which then decomposes in the presence of $H_2$ to form CO and $H_2O$.

$$CO_2(g) + H_2(g) \rightarrow COOH \cdot s + H \cdot s \rightarrow CO \cdot s + H_2O \cdot s \tag{5}$$

CO and $H_2O$ were the main products formed during gas-switching experiment (Figure 6). In the first three cycles, CO was formed when switching from $H_2$ to $CO_2$, and a very small amount of CO was formed when switching from $CO_2$ to $H_2$ at 30 min, 70 min and 110 min, respectively. When the catalyst was purged 20 min with helium before switching from $CO_2$ to $H_2$, CO was not formed, and $H_2O$ was produced at 195 min. Water was formed when switching from $H_2$ to $CO_2$ and when switching from $CO_2$ to $H_2$. After flowing $H_2$ and purging the reactor with He for 20 min, only a negligible amount of $H_2O$ was formed upon the admission of $CO_2$ (at 150 min).

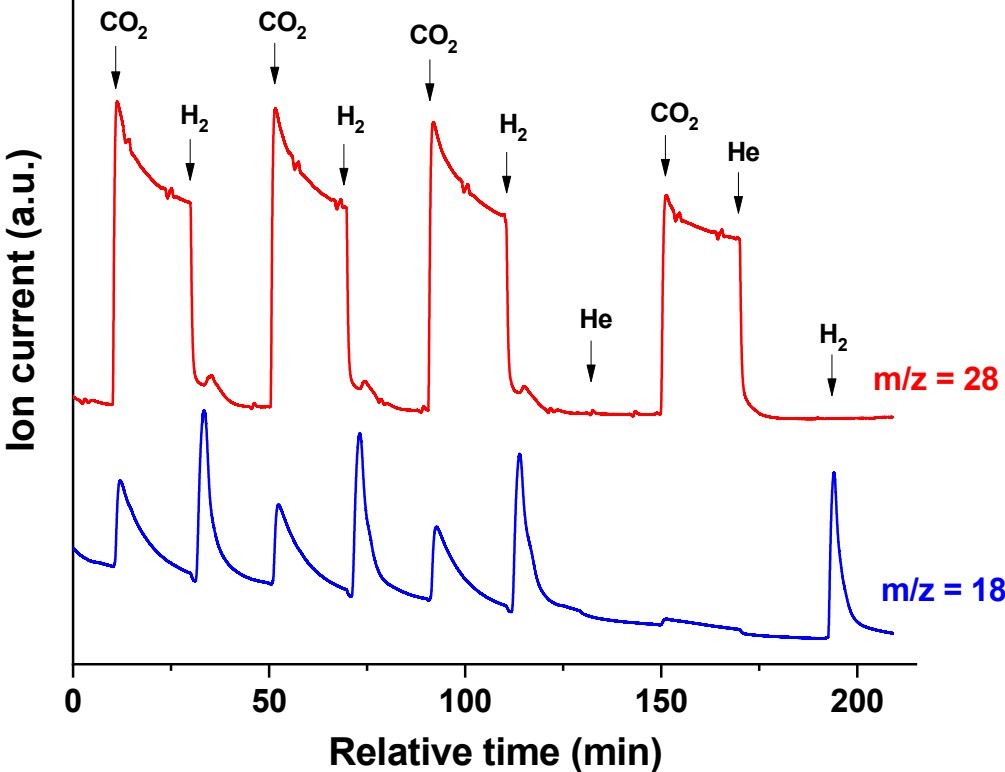

**Figure 6.** Ion current at m/z = 18 ($H_2O$) and 28 (CO) during $H_2$/$CO_2$ switching experiments on $Fe_3O_4$. Arrows with a label indicate a change in gas composition to the indicated gas. The catalyst first was reduced in flowing $H_2$ for 2 h, following the reaction in $H_2$ and $CO_2$ for 2 h, and $H_2$/He in 20 min before the first admission of $CO_2$. Reaction conditions: T = 773 K, $F_{He}$ = 36 sccm, $F_{H2}$ or $F_{CO2}$ = 4 sccm.

In Figure 6, the fact that CO was formed when the reduced form of the $Fe_3O_4$ catalyst was contacted with $CO_2$, (even after the purge with He to decrease the concentration of any surface $H_2$), is evidence of a redox pathway. During the first 125 min of gas-switching experiments, $H_2O$ was produced during flows of only $CO_2$ or only $H_2$. This differs from what is expected in the traditional redox cycle, in which $H_2O$ is only produced during the $H_2$ feeding period (see Equation (4)). However, after flowing $H_2$ and purging the reactor with He for 20 min, the admission of $CO_2$ only produced negligible amounts of water. Table 3 summarizes the estimated initial rates of CO production on the $Fe_3O_4$ catalyst during each segment of the gas-switching experiments. The CO production rates were calculated from the initial slopes of the concentration vs. time data in Figure 6, essentially modeling the system as a batch reactor (Equation (6)).

$$\frac{dC_A}{dt} = r_A \tag{6}$$

**Table 3.** Estimated initial rates of CO production after gas switches from $H_2$ to $CO_2$ and from $CO_2$ to $H_2$ during gas-switching experiment on $Fe_3O_4$ in Figure 6.

| Period | Rate after $H_2$ to $CO_2$ Gas Switch ($\mu mol\ L^{-1}\ s^{-1}\ g_{cat.}^{-1}$) | Rate after $CO_2$ to $H_2$ Gas Switch ($\mu mol\ L^{-1}\ s^{-1}\ g_{cat.}^{-1}$) | ($H_2$ to $CO_2$ Rate)/($CO_2$ to $H_2$ Rate) Ratio |
|---|---|---|---|
| 1st $CO_2$ | 2.78 | 1.64 | 1.70 |
| 2nd $CO_2$ | 2.98 | 1.00 | 2.98 |
| 3rd $CO_2$ | 2.74 | 0.91 | 3.01 |
| 4th $CO_2$ (after He purge) | 1.94 | 0 | - |

It is observed (Table 3) that the rate after switch from $H_2$ to $CO_2$ fluctuated between 2.74 $\mu mol\ L^{-1}\ s^{-1}\ g_{cat.}^{-1}$ and 2.98 $\mu mol\ L^{-1}\ s^{-1}\ g_{cat.}^{-1}$ in the first three periods of $CO_2$ admission, and it decreased after the He purge. The rate after switching from $CO_2$ to $H_2$ decreased in the first three periods, and it was zero (with no CO produced) during the last admission of $CO_2$ after the He purge. The decrease of the CO initial rate after switching from $CO_2$ to $H_2$, especially when equal to zero after the purge, raises doubts about the existence of residual $CO_2$ during the first three admissions of $H_2$ in the switching experiment. As a control, when the gas was switched from $CO_2$ to $H_2$, the $CO_2$ gas did not exit from the surface very quickly (see Figure S2). Therefore, a small amount of CO can be produced by the residual $CO_2$ with the available reduced sites; evidence of this interpretation in the detection of very small peaks after the $H_2$ admissions (Figure 6). The negligible CO production (relative time = 34 min, 74 min, and 114 min, respectively) after the $H_2$ admissions should not be considered evidence of the associative mechanism. In summary, the CO formation upon switching from $H_2$ to $CO_2$ is evidence consistent with the redox mechanism, while the small contribution of CO production upon switching from $CO_2$ to $H_2$ was suppressed by the confirmation of the helium purge. Thus, from the gas-switching experiment, only the redox pathway is active on our $Fe_3O_4$-derived catalyst.

Isotopic experiments were conducted to gain insight into the mechanism of the reaction. The isotopic $C^{18}O_2$ to the $CO_2$ switching experiment is shown in Figure 7. Here it can be seen that $C^{18}O$ (m/z = 30) formed and CO (m/z = 28) decreased when the gas ($CO_2/H_2$) was switched to $C^{18}O_2/H_2$. CO can be only formed from $CO_2$, and not from the lattice oxygen.

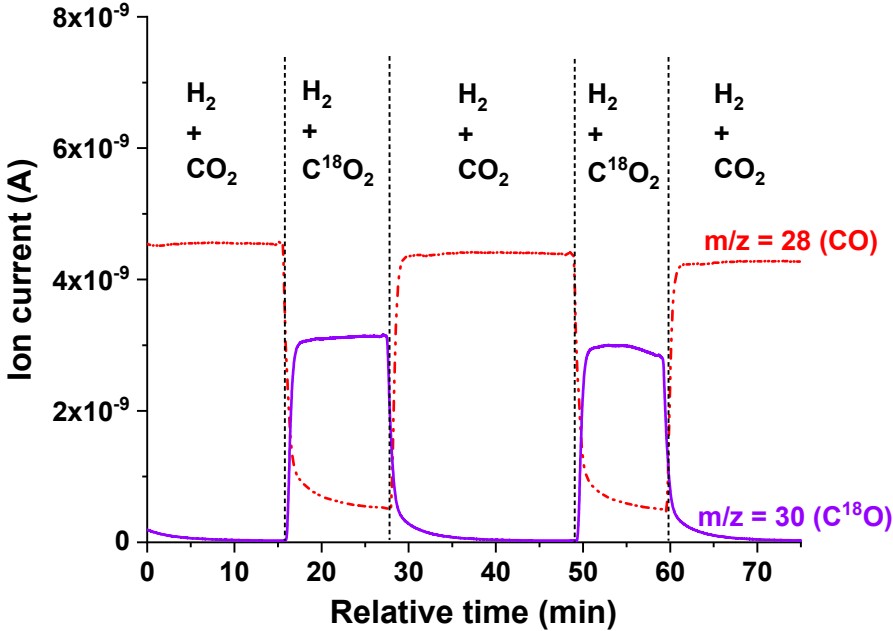

**Figure 7.** Ion current at m/z = 28 (CO) and 30 ($C^{18}O$) during $CO_2/C^{18}O_2$ switching experiments on $Fe_3O_4$ (100 mg). Labels in each region indicate a change in gas composition. A standard pretreatment (reduced in flowing $H_2$ for 2 h) and RWGS reaction (>2 h) had been done before the switching experiment. Switching experiment conditions: T = 753 K, $F_{tot}$ = 40 sccm, $F_{H2}$ = 4 sccm, $F_{CO2}$ or $F_{C18O2}$ = 4 sccm.

The gas-switching experiments with $CO_2$ and $H_2$ led us to conclude that a redox pathway is active on the $Fe_3O_4$-derived catalyst. A model that can present a redox reaction pathway for this catalyst is given in Scheme 1. It includes the adsorption of both of the reactants, $CO_2$ and $H_2$. In the surface redox mechanism, the dissociation of $CO_2$ at the catalyst surface (step 2 in Scheme 1) is known to be the RDS [8,30,34]. Evidence for $H_2$ dissociation (step 5 in Scheme 1) was observed when $H_2/D_2$ mixtures were fed to the catalyst in the presence of $CO_2$ (see Figure S3).

$$1)\ CO_{2\,(g)} + s \rightleftharpoons CO_2 \cdot s$$

$$2)\ CO_2 \cdot s + s \longrightarrow CO \cdot s + O \cdot s$$

$$3)\ CO \cdot s \rightleftharpoons CO_{(g)} + s$$

$$4)\ H_{2\,(g)} + s \rightleftharpoons H_2 \cdot s$$

$$5)\ H_2 \cdot s + s \rightleftharpoons 2\,H \cdot s$$

$$6)\ 2\,H \cdot s + O \cdot s \rightleftharpoons H_2O \cdot s$$

$$7)\ H_2O \cdot s \rightleftharpoons H_2O_{(g)} + s$$

**Scheme 1.** Redox reaction pathway for CO formation.

HD formation was observed to occur quickly, since the amount of $CO_2$ to CO conversion decreased on the same time scale when switching the concentration from $H_2/D_2$ (7.5 kPa/7.5 kPa) to $H_2$ (7.5 kPa), indicating that $H_2$ dissociation is reversible and not rate limiting.

An additional gas switching experiment was conducted to monitor the $H_2O$ production (Figure S4). The amount of $H_2O$ produced during $H_2$ flow periods was consistent between each cycle (Figure S4a),

that is, the adsorbed O·s species formed upon $CO_2$ reduction are stable at these reaction conditions. However, the amount of $H_2O$ produced during the period of $CO_2$ flow decreased as the purge time in helium increased. After only a five min purge, the amount of $H_2O$ produced during the period of $CO_2$ flow was much greater than that produced following a 20 min purge in helium, and the rate fitted from the initial slope of this region dropped dramatically (see Table S1). This suggests that H* atoms from the catalyst surface appeared to desorb (as $H_2$) during the He purge. This was a slow process, because even following a 20 min purge, there were enough H* atoms on the sample to form small amounts of $H_2O$ when $CO_2$ was administered. Both gas-switching experiments (Figure 6 and Figure S4) suggest that the redox mechanism should be the dominant reaction pathway for this $Fe_3O_4$-derived catalyst during the $CO_2$ hydrogenation at our reaction conditions (753 K, 1 atm).

## 3. Materials and Methods

### 3.1. Materials

Magnetite ($Fe_3O_4$, 99.99%, 50–100 nm particle size, Sigma-Aldrich, St. Louis, MO, USA) powder was pressed by hydraulic press and sieved (mesh 40 to mesh 60) to obtain particle sizes within the range of 250–425 μm before the reaction. Iron carbide ($Fe_3C$, 99.5%, American Element, Los Angeles, CA, USA) granules were crushed to a powder using a ball-mill, pressed using a hydraulic press, and sieved within the range of 250–425 μm. The gases used were $CO_2$ (Grade 5.0, Keen, Wilmington, DE, USA), $H_2$ (UHP, Matheson, Basking Ridge, NJ, USA), Helium (Grade 5.0, Keen, Wilmington, DE, USA), $C^{18}O_2$ (95 atom% $^{18}O$, Sigma-Aldrich, St. Louis, MO, USA), and $D_2$ ( 99.6% gas purity, 99.8% isotope purity, Cambridge Isotopes, Tewksbury, MA, USA).

### 3.2. Reactor Setup for Kinetics and Gas-Switching Experiments

The reaction rates and other kinetic parameters were measured using a packed-bed microreactor operated in a down-flow mode at atmospheric pressure. The catalyst particles were supported on a plug of quartz wool within a 7 mm I.D. quartz tube reactor. The quartz tube was positioned inside an Omega CRFC-26/120-A ceramic radiant full cylinder heater. The temperature was controlled by an Omega CN/74000 temperature controller using the input from a K-type, 1/16 in. diameter thermocouple (Omega Engineering, Inc., Norwalk, CT, USA) placed around the outside of the quartz tube at the center of the catalyst bed. Gas flows were controlled by mass flow controllers (Brooks Instruments) through the reactor or the other instruments.

Gas transfer lines were heated to a temperature above 373 K at all times to avoid water condensation. The composition of the effluent stream was analyzed online by a gas chromatograph (GC, 7890A, Agilent, Santa Clara, CA, USA) during continuous flow experiments or by a mass spectrometer (MS, GSD320, Pfeiffer Vacuum Technology AG, Aßlar, Germany) during gas-switching experiments or isotopic experiments. The GC was equipped with both a thermal conductivity detector (TCD) and a flame-ionization detector (FID). The TCD was used to quantify $CO_2$, CO and $H_2$ concentrations, while the FID was used to quantify hydrocarbon concentrations. An Agilent 2 mm ID × 12 ft Hayesep Q column was used in the GC to separate products quantified with the TCD, and an Agilent 0.32 mm ID × 30 m HP-Plot Q column was used to separate products quantified with the FID.

### 3.3. Catalyst Characterization

Temperature-programmed reduction (TPR) was performed by using a MS including the reduction period (773 K, $P_{tot}$ = 1 bar, $P_{H2}$ = 15 kPa in He for 2 h), and then the temperature was ramped up to 1073 K. The ramping rate of each step was 5 K min$^{-1}$. Another TPR experiment was performed after the reverse water-gas shift (RWGS) reaction (553K, $P_{tot}$ = 1 bar, GHSV = 4.5 × 10$^4$ cm$^3$ h$^{-1}$ g$_{cat.}$$^{-1}$, $P_{CO2}$ = $P_{H2}$ = 15 kPa with He as an inert balance gas). The post-reaction sample was cooled down to room temperature in He flow inside the microreactor. The TPR was then continued in 15 kPa $H_2$ and 86.3 kPa He from room temperature to 1073 K at the rate of 5 K min$^{-1}$. The $H_2$ consumption was

converted from the ion current (m/z = 2) after the calibration of the $H_2$ signal each time before the experiment using a mass spectrometer.

X-Ray Diffraction (XRD) patterns of catalyst powders were collected at room temperature on a Bruker diffractometer using Cu K$\alpha$ radiation ($\lambda$ = 1.5418 Å). Measurements were taken over the range of $5° < 2\theta < 70°$, with a step size of 0.02° before and after the RWGS reaction. X-ray photoelectron spectroscopy (XPS) measurements were performed on a K-alpha Thermo Fisher Scientific spectrometer using a monochromated Al K$\alpha$ X-ray source. The measurements of iron oxide samples (pre- and post-reaction) were done with a spot size of 50 μm at ambient temperature and a chamber pressure of ~$10^{-7}$ mbar. A flood gun was used for charge compensation. All the spectra measured were calibrated by setting the reference binding energy of carbon 1s at 284.8 eV. The spectra were analyzed by Thermo Fisher Scientific (Waltham, MA, USA) Avantage® commercial software (v5.986). For the fitting, each component consists of a linear combination of Gaussian and Lorentzian product functions, and the full width at half maximum (FWHM) and differences in binding energy of the same species between the Fe2p$_{3/2}$ and Fe2p$_{1/2}$ scan were kept constant. SMART background in Avantage® was used over the region to define the peaks.

### 3.4. Measurement of Product Formation Rates and Reaction Rates with Gas-switching or Isotopic Experiments

Most of the procedures in the measurement of the reaction rates, gas-switching, or isotopic experiments are identical to the ones described in our previous report [14]. $Fe_3O_4$ samples were pretreated before all experiments by increasing the reactor temperature at a rate of 5 K min$^{-1}$ to 773 K under a gas flow of 15 kPa $H_2$. After the pretreatment at 773 K for 2 h, the temperature was lowered to the initial reaction temperature of 753 K. During the measurement of the reaction rates, the partial pressures of the reactants were $P_{CO2}$ = $P_{H2}$ = 15 kPa. A constant total flow rate of 75 cm$^3$ min$^{-1}$ (sccm) was maintained with He as an inert balance gas.

Gas hourly space velocity (GHSV) in cm$^3$ h$^{-1}$ g$_{cat.}$$^{-1}$ was calculated under STP condition (273 K and 1 atm) according to the equation below:

$$GHSV = \frac{F_{tot}}{m_{cat.}} \tag{7}$$

where $m_{cat.}$ is the mass of the catalyst and $F_{tot}$ is the total flow rate.

$CO_2$ and $H_2$ reaction orders were measured by independently varying the inlet $CO_2$ and $H_2$ partial pressures. The total pressure remained at 1 bar. The activation energy was estimated by using the Arrhenius plot with the temperature varied between 723 K and 773 K while monitoring the CO formation rate.

Rates of CO formation were calculated assuming differential reactor operation according to Equation (8):

$$r_{CO} = \frac{\dot{V}\Delta C_{CO}}{m_{cat.}} \tag{8}$$

where $\dot{V}$ is the total volumetric flow rate, $\Delta C_{CO}$ is the change in CO concentration. Measured reaction rates are the net rate of the forward and reverse reactions; therefore, the observed rate must be transformed into the reaction rate for the forward reaction by using Equations (9)–(11). The equilibrium constant ($K_C$) is low (<1) for the RWGS at the temperatures investigated, although the reverse reaction had a negligible contribution to the observed rates because of the low conversion (<12%) under conditions at which the reactor was operated. Note that $C_o$ (Equation (11)) represents the standard state (1 mol L$^{-1}$) and equals 1, since the reaction is equimolar.

$$r_{obs.} = r_+ - r_- = r_+(1 - \eta) \tag{9}$$

$$\eta = \frac{[CO][H_2O]}{K_C[CO_2][H_2]} \tag{10}$$

$$K_C = \left( \prod_i C_{i_{eq.}}^{\gamma} \right) \frac{1}{C_o} \tag{11}$$

where $r_{obs.}$ is the observed rate, $r_+$ and $r_-$ are the rate of forward and reverse reactions, $\eta$ is the ratio of the rate of the reverse and forward reactions and $K_C$ is the equilibrium constant.

Experiments were also conducted to *(i)* determine reaction rates in excess (i.e. non-equimolar) $CO_2$ or $H_2$, and *(ii)* to determine apparent kinetic parameters. In the first case, $CO_2$ and $H_2$ were fed with the catalyst—$Fe_3O_4$ (100 mg)—held at a temperature of 753 K. The initial partial pressure of both $CO_2$ and $H_2$ was 15 kPa. After a period of 16 h, the partial pressure of $CO_2$ was increased to 60 kPa, while the partial pressure of $H_2$ was held at 15 kPa. After another period of 2 h, the partial pressure of $CO_2$ was decreased to 15 kPa and the partial pressure of $H_2$ was increased to 60 kPa. Finally, both partial pressures were returned to 15 kPa. $CO_2$ conversion was quantified under the same conditions.

For the second case, apparent kinetic parameters (activation energy and reaction orders) were determined with near equimolar concentrations of $CO_2$ and $H_2$ on $Fe_3O_4$, and under large $H_2$ excess on $Fe_3O_4$ as well. With near equimolar concentrations of $CO_2$ and $H_2$, the reaction was first performed for 15–16 h at a temperature of 753 K with reactant partial pressures of 15 kPa. The temperature was then lowered in 10 K increments to 723 K, with 5–6 GC injections (a period of about 60 min) taken at each temperature. After the period at 723 K, the $CO_2$ partial pressure was reduced to 10 kPa and increased in 2.5 kPa increments to a final partial pressure of 20 kPa. Finally, the $CO_2$ partial pressure was returned to 15 kPa and the $H_2$ partial pressure was lowered to 10 kPa and increased in 2.5 kPa increments. The basic outline of experiments conducted with excess $H_2$ was the same as that used for near equimolar reactant concentrations (see also our previous report) [14]. Reactant partial pressures during the initial period were 90 kPa $H_2$ and 10 kPa $CO_2$. During the variable $CO_2$ partial pressure period, the $H_2$ partial pressure was 85 kPa, and the $CO_2$ partial pressure was varied between 5 and 12.5 kPa in 2.5 kPa increments. To investigate the effect of $H_2$ partial pressure, the $CO_2$ partial pressure was kept at 10 kPa and the $H_2$ partial pressure was varied between 70–90 kPa in 5 kPa increments.

The measurement of the CO formation rate over $Fe_3C$ has been performed in two different manners. First, the temperature was ramped under a flow of $H_2$ (15 kPa) and He at the rate of 5 K min$^{-1}$ up to 753 K. 15 kPa $CO_2$ was then added into the flow while monitoring the CO formation rate. The second part of the $Fe_3C$ activity test was done by ramping the temperature to 573 K, 623 K, 673 K, 723 K and then cooling back to 673 K again at the rate of 5 K min$^{-1}$ under the flow of $H_2$ (15 kPa), $CO_2$ (15 kPa), and He (remainder). Each temperature was held for 80 min respectively for GC injection.

Gas-switching experiments were done by measuring CO formation rates while alternating between $CO_2$ and $H_2$ gas flows. Catalysts were pretreated as described above, and after pretreatment, $CO_2$ was added into the reactor to allow the RWGS reaction to proceed for 2 h.

After the reaction, the gas flow rates were changed to 36 sccm helium and 4 sccm $H_2$. After 20 min, $H_2$ flow was stopped and was replaced by 4 sccm of $CO_2$. After 20 min, $CO_2$ in the gas stream was replaced by $H_2$. This $CO_2 \rightarrow H_2$ sequence was repeated three times. The reactor was then purged with helium for 20 min before $CO_2$ was readmitted into the gas stream. After 20 min, the reactor was again purged with helium before $H_2$ was readmitted to the gas stream. All sequences with a given gas composition lasted for 20 min, and the temperature of the reactor was 773 K throughout the duration of the gas switching portion of the experiment.

Additional gas-switching experiments involving purge times of varying length with an inert gas were carried out at 753 K. Following the same pretreatment and the reaction in 15 kPa $H_2$ and 15 kPa for 2 h at 773 K, 15 kPa $H_2$ was admitted to the reactor. After 15 min, $H_2$ was replaced by 15 kPa $CO_2$ for 15 min, and $CO_2$ was then replaced by $H_2$ for another 15 min. Then the reactor was purged with helium for 5 min. This sequence ($CO_2 \rightarrow H_2 \rightarrow$ He) was repeated several times, but each time the length of the inert purge was increased by 5 min.

An isotopic experiment for the CO formation rate was monitored by MS on $Fe_3O_4$ (100 mg) while alternating between $CO_2$ (4 sccm) and $C^{18}O_2$ (4 sccm) after the RWGS reaction for 2 h. The temperature was kept constant at 753 K. The total flow rate was 40 sccm with $H_2$ maintained at 4 sccm.

The kinetic isotope effect (KIE) of $H_2/D_2$ was investigated on $Fe_3O_4$ (100 mg) for various $H_2:CO_2$ ratios. After pretreatment, the reaction began at a temperature of 753 K with $CO_2$ and $H_2$ partial pressures of 15 kPa. After 16 h, the temperature was lowered to 723 K, and after 1.5 h, $H_2$ in the feed was replaced by $D_2$.

## 4. Conclusions

An unsupported $Fe_3O_4$-derived catalyst showed very promising activity toward CO formation via $CO_2$ hydrogenation. The high selectivity (~100% under $H_2:CO_2$ = 1:1) and great stability make the catalyst feasible to consider in extensive use. The catalyst exhibited only slight deactivation under conditions of excess $CO_2$, but it can be quickly regenerated under excess $H_2$. Reaction rates depended more strongly on $H_2$ (0.8 in reaction order) compared to $CO_2$ (0.33 in reaction order) under a near equimolar gas-phase composition. The post-reaction analyses of the catalyst indicated that the catalyst was reduced to metallic iron first in the pretreatment of $H_2$, but the working catalyst remained partially oxidized with the composition of $Fe^{2+}$, $Fe^{3+}$, $Fe^0$ and $Fe_3C$. The main active sites are believed to be the combination of the above species, except that $Fe_3C$ is unlikely to directly contribute to the very steady CO formation at our reaction conditions (1 atm, 723–773 K). Gas-switching experiments revealed that CO was formed only when switching from $H_2$ to $CO_2$, and $H_2O$ was formed when switching from $CO_2$ to $H_2$, but not when switching from $H_2$ to $CO_2$ if purging of helium was in between with the gas admission. The redox mechanism is identified as the dominant reaction pathway for the unsupported iron catalyst.

**Supplementary Materials:** The following are available online at http://www.mdpi.com/2073-4344/9/9/773/s1, Figure S1. XRD pattern of $Fe_3O_4$ after the reduction in $H_2$, Figure S2. Ion current at m/z = 18 ($H_2O$), 28 (CO), and 44 ($CO_2$) during $H_2/CO_2$ switching experiments on $Fe_3O_4$. Arrows with a label indicate a change in gas composition to the indicated gas. Reaction conditions: T = 773 K, $F_{He}$ = 36 sccm, $F_{H2}$ or $F_{CO2}$ = 4 sccm. The figure is a modification of Figure 6, Figure S3. Ion current at m/z = 2 ($H_2$), 3 (HD), 4 ($D_2$), and 28 (CO) during flow of 7.5 kPa $H_2$ + 7.5 kPa $D_2$ +15 kPa $CO_2$ and 7.5 kPa $H_2$ +15 kPa $CO_2$ on $Fe_3O_4$. Reaction conditions: T = 753 K, $F_{tot}$ = 75 sccm, Figure S4. (a) Ion current at m/z = 28 (CO) and m/z = 18 ($H_2O$) during $H_2/CO_2$ switching experiments on $Fe_3O_4$. Arrows with a label indicate a change in gas composition to the indicated gas. The catalysts were in flowing $H_2$ for 2 h followed by the reaction in $CO_2$+$H_2$ for 2 h before the first admission of $H_2$ (relative time: 31 min) and $CO_2$ (relative time: 46 min). Reaction conditions: T = 753 K, $F_{tot}$ = 75 sccm, $P_{H2}$ or $P_{CO2}$ = 15 kPa. (b) is the modification of (a), Figure S5. XPS O1s spectra of $Fe_3O_4$ and post-reaction $Fe_3O_4$. The curves under the fitted envelope and above the background are contributions of estimated components from peak fitting. The peak deconvolution and their atomic % are listed on the right of the spectra for each sample, Table S1 Fitted initial slopes and area of $H_2O$ in $H_2/CO_2$ switching experiment with different He purging time in Figure S4

**Author Contributions:** Conceptualization, C.-Y.C. and R.F.L.; methodology, C.-Y.C., J.A.L., and R.F.L.; investigation, C.-Y.C. and R.F.L.; resources, R.F.L.; data curation, C.-Y.C.; writing—original draft preparation, C.-Y.C.; writing—review and editing, C.-Y.C., J.A.L., and R.F.L.; supervision, R.F.L.; project administration, R.F.L.; funding acquisition, R.F.L."

**Funding:** This research was funded by U.S. Army, grant number GTS-S-17-013.

**Acknowledgments:** The authors would also like to acknowledge Terry Dubois for the help and suggestions.

**Conflicts of Interest:** The authors declare no conflict of interest.

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
