# Peer review of "Reverse Water-Gas Shift Iron Catalyst Derived from Magnetite"

_catalysts, doi:10.3390/catal9090773_

Round 1

Reviewer 1 Report

The current version of manuscript submitted by Lobo and co-workers presents a comprehensive study on iron-catalyzed RWGS reaction. The quality of the work fulfills the expectation of manuscripts that can be published on the journal of Catalysts. The only suggestions for further improvement are 1) Moessbauer spectrometry might make the manuscript more productive; 2) a global kinetic expression (time-dependent) might be preferred including all possible active species, even though the iron-carbide does not contribute to the steady-state rate. But both points are not required, the manuscript can be accepted as the present version.

Author Response

We appreciate the comments of the reviewer. Unfortunately we are not able to carry out the experiments suggested because we do not have timely access to a Moessbauer spectrometer. Similarly, a global kinetic expression would require many more measurements than the ones reported here and is not possible within a reasonable amount of time. This is a good suggestion that we hope we can fulfill in a future study. 

Reviewer 2 Report

The authors performed a decent piece of work on catalyst testing for the reverse WGS reaction. Since their magnetite catalyst shows a high activity and stability it is of interest for industry. The paper is of high quality and deserves to be published after some minor amendments which follow below.

Figure 1: The authors call a stable performance during about 600 minutes already a proof of a high stability of the catalyst. In industry, however, the catalyst has to last a few years, which is at least three orders of magnitude more. The authors should therefore relativate their claim of the high catalyst stability.

Section 3.1: Please add some details about how the magnetite catalyst was prepared. Specification of the starting powder, way of pressing, way of crushing and sieving (by mortar and by hand ?).

Section 3.2: Please check for absence of significant transport limitations, such as the Wheeler-Weiss modulus and radial heat transport limitation in the bed (e.g. see: Mears D.E., Diagnostic criteria for heat transport limitations in fixed bed reactors, J. Catal. 20 (1971) 127-131). I expect that both limitations are significant but not very large, so the conclusions will probably not be affected.

Line 409: The GHSV is expressed in cm3 h-1 gcat-1. Please specify explicitly that cm3 is a STP (0°C and 1.0 atm. (or whatever was used)).

Line 418 (Eq. (8)): at a CO2 conversion of 33.7 % the differential approach for calculating the reaction rate yields a rather large deviation. Better would be the integral approach, but due to the complexity in the reaction orders (both for H2 and CO2 and maybe the orders also depend on the conversion) this is rather awkward. Therefore I agree that the differential approach is preferred, but a small remark of the deviation at higher conversion should be added.

Equation (9-10-11): all the symbols used should be explained explicitly.

Author Response

The authors performed a decent piece of work on catalyst testing for the reverse WGS reaction. Since their magnetite catalyst shows a high activity and stability it is of interest for industry. The paper is of high quality and deserves to be published after some minor amendments which follow below.

Figure 1: The authors call a stable performance during about 600 minutes already a proof of a high stability of the catalyst. In industry, however, the catalyst has to last a few years, which is at least three orders of magnitude more. The authors should therefore relativate their claim of the high catalyst stability.

Response:

Our catalysts showed high stability based on the fact that not only the deactivation “rate” is negligible under the condition of H2:CO2 = 1:1, but also the possibility of reactivation rate with higher concentration of H2 flow (see the condition of H2:CO2= 4:1 in Figure 1). The description of the stability in Figure 1 is modified as follows:

The catalyst showed overall high stability in 1300 min, and the final reactivation rate in excess H2 was higher than the deactivation rate under excess CO2 condition. This indicates that this Fe3O4-derived catalyst is easy to regenerate in a very short period (< 100 min), making the catalyst potentially attractive in industrial use for long-term application.

Section 3.1: Please add some details about how the magnetite catalyst was prepared. Specification of the starting powder, way of pressing, way of crushing and sieving (by mortar and by hand ?).

Response:

The description of the catalyst preparation is revised to the following sentences in the Materials and Methods section:

Magnetite (Fe3O4, Aldrich, 99.99%, 50-100 nm particle size) powder was pressed by hydraulic press machine and sieved (mesh 40 to mesh 60) to obtain particle sizes within the range of 250-425μm before the reaction. Iron carbide (Fe3C, American Element, 99.5%) granules was crushed to powder by a ball-mill, pressed by hydraulic press machine, and sieved within the range of 250-425μm.

Section 3.2: Please check for absence of significant transport limitations, such as the Wheeler-Weiss modulus and radial heat transport limitation in the bed (e.g. see: Mears D.E., Diagnostic criteria for heat transport limitations in fixed bed reactors, J. Catal. 20 (1971) 127-131). I expect that both limitations are significant but not very large, so the conclusions will probably not be affected.

Line 409: The GHSV is expressed in cm3 h-1 gcat-1. Please specify explicitly that cm3 is a STP (0°C and 1.0 atm. (or whatever was used)).

Response:

The definition of GHSV in the manuscript is specified that it is a STP conditions. The description is modified to:

Gas hourly space velocity (GHSV) in cm3 h-1 gcat. -1 was calculated under STP condition (273 K and 1 atm) according to the equation below:

Where  is the mass of the catalyst and Ftot is the total flow rate.

Line 418 (Eq. (8)): at a CO2 conversion of 33.7 % the differential approach for calculating the reaction rate yields a rather large deviation. Better would be the integral approach, but due to the complexity in the reaction orders (both for H2 and CO2 and maybe the orders also depend on the conversion) this is rather awkward. Therefore I agree that the differential approach is preferred, but a small remark of the deviation at higher conversion should be added.

Response:

We thank the reviewer for pointing out that a deviation of reaction rate at higher CO2 conversion could happen. A remark is added to the text in the discussion of Figure 1 as follows:

It should be noted that the differential condition (see Eq. (8)) was used to determine reaction rates, and preferred for the investigation of the kinetic properties of our materials. Under higher concentration of H2 (15 kPa CO2 + 60 kPa H2), high CO2 conversion (>12%) could lead to small errors in the estimation of reaction rate and the reaction order. The trend observed in Figure 1, however, should not be affected by this approximation.

Equation (9-10-11): all the symbols used should be explained explicitly.

Response:

The symbols are further specified after Eq. 9-11. The changes are as follows:

where robs is the observed rate, r+  and r- are the rate of forward and reverse reactions,  is the ratio of the rate of the reverse and forward reactions,  is the equilibrium constant.

Reviewer 3 Report

The authors investigated the catalytic properties of magnetite microparticles (250-425 µm) towards the reverse water-gas shift reaction at elevated temperatures (723-773 K).

The manuscript is well planned and thought out. The experimental procedures are described adequately. The introduction leads to the discussion and the references are balanced.

There are only a few minor points:

The textual quality of Equation 2 and Scheme 1 should be improved.

The multiplett splitting for the interpretation of XP spectra should be considered according to the references (Biesinger et al 2011 and McIntyre 1977). Fe2p3/2 spectra are quite complex and especially the fitting of Fe2+ components is quite challenging. Since you try to evaluate the share of different iron species, I would recommend to fit these multipletts in order to obtain correct shares of the iron species.

Author Response

Response:

The quality of Eq. 2 and Scheme 1 is now improved.

The multiplet splitting of Fe2P3/2  spectra was evaluated and was referred to the following references during the data analysis.

[40] Yamashita, T.; Hayes, P. Analysis of XPS spectra of Fe2+ and Fe3+ ions in oxide materials. Applied Surface Science 2008, 254, 2441–2449.

[41] Biesinger, M.C.; Payne, B.P.; Grosvenor, A.P.; Lau, L.W.M.; Gerson, A.R.; Smart, R.S.C. Resolving surface chemical states in XPS analysis of first row transition metals, oxides and hydroxides: Cr, Mn, Fe, Co and Ni. Applied Surface Science 2011, 257, 2717–2730.

[42] McIntyre, N.S.; Zetaruk, D.G. X-ray photoelectron spectroscopic studies of iron oxides - Analytical Chemistry (ACS Publications). Analytical Chemistry 1977, 49, 1521–1529.

To keep the results more straightforward, the authors chose to combine several multiplet that represent to same oxidation state to one peak, which did not change the overall concentration of each species that we would like to investigate.

The fitting was done carefully and combined with the scientific understanding. The authors believe that the results are reliable and are shown in Table 2.